# Exploring Context Allows Us to Better Understand Physical Activity in People with and Without Parkinson’s Who Have Fallen: A Mixed Methods Study

**DOI:** 10.3390/geriatrics10010008

**Published:** 2025-01-07

**Authors:** Katherine Baker, Julia Das, Lynn Rochester, Silvia Del Din, Jenni Naisby

**Affiliations:** 1Department of Sport, Exercise & Rehabilitation, Northumbria University, Newcastle upon Tyne NE7 7XA, UK; julia.das@northumbria.ac.uk (J.D.);; 2Translational and Clinical Research Institute, Newcastle University, Newcastle upon Tyne NE4 5PL, UKsilvia.del-din@newcastle.ac.uk (S.D.D.); 3National Institute for Health and Care Research (NIHR), Newcastle Biomedical Research Centre (BRC), Newcastle University and The Newcastle upon Tyne Hospitals NHS Foundation Trust, Newcastle upon Tyne NE4 5PL, UK

**Keywords:** Parkinson’s, falls, physical activity, exercise, mixed methods

## Abstract

**Background:** Falls are a frequent and serious problem for older adults, especially for those living with Parkinson’s. The relationship between falls and physical activity is complex, and people often restrict activity following a fall. Exercise is an important aspect of reducing further risk of a fall and a key component of the management of Parkinson’s. The aim of this study was to understand the types of activity they are engaged in, the environments in which they take place, and the experience of people with and without Parkinson’s who have fallen. **Method**: Seventeen people with Parkinson’s and thirteen older adults who had experienced at least one fall in the previous year were recruited to this mixed methods study. Activity levels were captured over one week using accelerometers and body-worn cameras, allowing the type and location of activity to be recorded and analysed. This information informed an interview. **Results**: Findings showed that although both groups often achieved up to 10,000 steps per day, this was in very short bouts of activity. Sedentary activity, such as watching television, dominated the findings. Participants were aware of the benefits of being active but described many barriers to achieving the level of activity they would like to.

## 1. Introduction

Falls are a frequent and serious problem for many people with Parkinson’s (PwP), contributing to reduced quality of life and mobility [1]. On average, people with Parkinson’s fall at least once per year [2]. Primary disease processes, including motor symptoms such as bradykinesia and postural instability leading to changes in gait and balance, cognitive dysfunction and affective disorders, can all contribute to falls. Falls risk is further increased by secondary complications due to deconditioning (muscle weakness and inactivity) or diminished confidence and fear of falling [3]. There are many generic and Parkinson’s-specific, as well as intrinsic and extrinsic, factors that potentially contribute to falls risk [1,4].

There is a complex association between falls and physical activity (PA) in PwP, with fallers more likely to lead a sedentary lifestyle [5]. Falls are less frequent in early disease stages when symptoms are more subtle and activity levels tend to be less affected, although it has been shown that less active individuals are more likely to fall in this early stage [6]. Falls risk increases with disease progression and severity, but as symptoms become more complex in later stages, and perhaps due to the loss of confidence seen after a fall, activity levels often reduce with an associated reduction in falls risk [7]. This creates tension for professionals who promote exercise, as remaining physically active is an important factor in managing motor and non-motor complications of Parkinson’s [6]. The nature of the fall varies considerably for those in the early stage of the disease, who are active and mobile but have diminished postural control and, therefore, increased exposure to more challenging activities, compared to those in the later stage of the disease, who have more limited mobility. As the disease progresses, there may initially be a reduction in falls due to a restriction of movement and, therefore, risk, but this eventually leads to a further increase in falls as individuals become deconditioned and fall during lower-level activities such as transfers [8,9]. Falls are not exclusively associated with moderate or severe disease; a study examining pre-fall events and activity levels in a group of early-stage PwP found that falls are indeed common and associated with reduced levels of ambulatory activity [6]. The study emphasises the importance of understanding the context of the fall (for example, during transitions or while walking) in targeting appropriate management strategies.

Despite the wealth of literature related to falls and physical activity in older adults and PwP, there are comparatively fewer qualitative studies exploring balance and fall perceptions in PwP [10]. Jonasson et al. (2019) conducted interviews with 12 PwP to explore their fear of falling [11]. The authors concluded that PwP’s perceptions in relation to falls and physical activities were complex and varied in relation to different activities and environments. Wearable technology provides a promising approach to understanding ambulatory behaviour and changes in movement patterns and quality outside of the clinic or laboratory [12]. Macro and micro gait characteristics have shown an association between fall history, activity pattern and variability of walking bouts, which enhance our understanding of falls risk [13]. We hoped to extend this understanding by exploring the context within which the PA and ambulation were taking place in order to further understand the complexity of falls in PwP.

Current guidance for addressing falls emphasises the importance of a personalised and multi-dimensional model [14]. It is important to acknowledge that simply advising an increase in physical activity may actually put people at increased risk of falls, meaning that it is important that any exercise or physical activity-based intervention is designed to address the complex picture associated with falls in Parkinson’s [12]. Whilst the importance of exercise is increasingly recognised due to the benefits for motor and non-motor symptoms [15,16], it remains challenging for many PwP to engage with and/or maintain PA due to a range of barriers, including physical discomfort, low self-efficacy and lack of social support [17].

The relationship between falls and PA warrants further investigation. Having more insight into the types of activities PwP engage with and the environments in which they spend time will allow us to better tailor our advice around maintaining PA while reducing falls risk. The aim of this study, therefore, was to explore the types of activities and environments of PwP who have fallen to identify barriers and facilitators to PA.

## 2. Materials and Methods

### 2.1. Design and Ethical Considerations

A mixed methods study design incorporating assessment of gait and balance, quantitative measurement of community-based activity using activity monitors, body-worn cameras and semi-structured interviews was employed to gain a detailed and rich understanding of factors relating to PA and falls. An explanatory sequential design was used; quantitative data were collected and analysed, which then informed the collection of the qualitative data collection and analysis with a final stage of overall interpretation [18]. Mixing occurred at the data collection stage, where the quantitative measures and body-worn camera images were connected to the semi-structured interview questions [18]. Mixing also occurred at the data analysis stage through the merging of the data sets to facilitate comparison and interpretation [18]. PwP were consulted during the design of the research to confirm the relevance of the issues relating to falls and PA. Feedback was given on the acceptability of the design and appropriateness of information sheets. The research team also involved three PwP who sat on the project steering group.

Ethical approval was granted by the NRES Committee North East (ref 16/NE/0296). The main ethical issue arising from this study related to the use of body-worn cameras, which take images as the participant moves around in their day-to-day life and, therefore, have the potential to include images of individuals who have not consented to take part in the research. This issue has been addressed in previous studies, and literature on these specific ethical issues was used to inform the current study design [19,20]. The participants were encouraged to wear the camera at all times (with specific exceptions, for example, in the bathroom).

The privacy of other people in the images, known as ‘secondary participants’, was of great importance, and a number of steps were taken to minimise the intrusion into these individuals’ privacy. Firstly, the participants were informed in the Participant Information Sheet and verbally that they must remove the camera in specific places where sensitive images may be captured: GP’s office, schools or swimming pools. When other people entered the participant’s home or when they entered the homes of others, the participants were told to inform other people of the camera and offer to remove it if they prefer. A brief information card was provided to support this. The behaviour of other individuals was not analysed in this study.

### 2.2. Recruitment and Study Sample

Purposeful sampling methods were employed to recruit PwP and older adults without Parkinson’s (OA) through a local multidisciplinary Parkinson’s service and via adverts in relevant community groups.

Inclusion criteria for PwP were diagnosis of idiopathic PD, independently mobile with or without a walking aid, self-report of at least one fall in the previous 12 months, adequate sight, hearing and cognition to tolerate the initial assessment and interview, ability to operate the body-worn camera and living in their own home (not in a residential home or inpatient in hospital). Inclusion criteria for OA were as follows: independently mobile with or without a walking aid, self-report of at least one fall in the previous 12 months, adequate sight, hearing and cognition to tolerate the initial assessment and interview and living in their own home (not in a residential home or inpatient in hospital). Potential participants were excluded if they presented dementia or cognitive difficulty, which would limit the ability to provide informed consent and to participate fully in the study (complete the assessments, manage the equipment and tolerate the interview).

### 2.3. Data Collection

All data collection took place in the participant’s own home with a registered physiotherapist (with over 15 years of clinical experience and qualitative research training) (J.D.). After written informed consent was given, an initial assessment of walking and balance was carried out, including a 10-metre walk test to measure gait speed, the Timed Up and Go test [21], the Mini BESTest test [22] and the falls efficacy scale [23]. In addition, the geriatric depression scale [24] was used to indicate the presence and severity of depression. Participants with Parkinson’s were also assessed for disease severity with the Hoehn and Yahr scale [25], and severity of motor symptoms with the UPDRS Part III (motor subsection) [26] and the new freezing of gait questionnaire [27].

Participants were then asked to wear the Axivity (AV3) monitor (Axivity Ltd., Newcastle, UK) and an Autographer body-worn camera (OMG Life, Oxford, UK), which automatically takes photographs from the point of view of the person wearing the camera every 30 s, for 7 days. The Axivity monitor was secured in place on the lower back with tape; the participants were advised to leave it in place and provided with additional tape and instructions should it need to be replaced (see Figure 1a). The body-worn camera was worn on a lanyard around the neck and, therefore, captured forward-facing images from the mid-chest level (see Figure 1b). While participants were encouraged to wear the camera for as much of the 7-day period as possible, it was also made clear that they could pause the camera or simply conceal it under their clothes any time during the 7 days, and they were given a brief written explanation to show to other people to explain why they were wearing the camera.

The researcher returned 7 days later to collect the equipment, at which point the participants had the opportunity to remove any images they did not wish to be included.

Following preliminary analysis of the Axivity and camera data, the researcher returned for a third visit within one week of the recording to carry out a semi-structured interview with the participant. The data from both the Axivity monitor and camera provided an overview of the amount, type and location of activities the person had taken part in, and this was used to form the basis of the interview. Influences on daily routines and any circumstances that either limit or facilitate PA were explored. The interview explored the experience of falling and whether this was felt to influence PA. If any falls occurred during the recording period, the circumstances were discussed with the participant. In addition, if the participant had adapted their activity because they considered themselves to be at risk or had a fear of falling, the photographs were used to explore these issues.

### 2.4. Outcomes

#### 2.4.1. Activity Data

Activity data were captured using the validated Axivity (AV3) monitor, which is a small and lightweight (23 × 32.5 × 7.6 (mm), 11 g) accelerometer worn on the lower back (Figure 1a). Activity is recorded at a sampling frequency of 100 Hz (16-bit resolution) and at a range of ±8 g. Recorded signals are stored locally on the sensor’s internal memory (512 MB) as a raw binary file and downloaded upon the completion of each 7-day participant recording period. Raw data were uploaded, and the following outcomes were used to describe the volume, pattern and variability of ambulatory activity:Total walk time (h)Total walk time per day (min)Steps per dayMean bout lengthPercentage of walking time per day

#### 2.4.2. Body-Worn Camera Data

Images from the body-worn camera were uploaded to a viewing software (Version 1), with each image time stamped. The images were manually coded according to the type of activity and location (e.g., indoors—home, indoors—not home, outdoors or in vehicle) to provide important contextual information about the participant’s activity. These codes were then used to quantify the time spent in each category. In order to normalise the data against the amount of time the cameras were active for each participant, these codes are expressed as percentages of the overall number of images for each participant.

### 2.5. Data Analysis

#### 2.5.1. Quantitative Data

Demographic, clinical, accelerometry and camera data were described for those with and without Parkinson’s. After examining the distribution, between-group differences were compared using the paired *t*-test or the Mann–Whitney test when assumptions for parametric testing were not met.

#### 2.5.2. Interview Data

Interviews were transcribed verbatim. Data were anonymised at transcription. The framework approach was used to guide the analysis [28,29]. Framework analysis consists of a matrix-based method that assists with the ordering and synthesising of qualitative data [30]. This approach sits broadly within the thematic analysis [31]. The framework approach involves interrelated steps, which include familiarisation, identifying a thematic framework, applying this framework through indexing, charting and mapping and interpretation [28,32]. Due to the series of stages, framework analysis is considered to be credible, as it demonstrates a clear audit trail of the steps of data analysis and how the raw data becomes the final presentation of findings [32]. Two members of the research team (J.N. and J.D.), who are both physiotherapists, were involved with the data analysis process. These members independently familiarised themselves with the data through reading transcripts, listening to audio recordings, and developing a coding framework. Initial codes were developed to begin the construction of a thematic framework. This was performed independently with the first three transcripts, and then the researchers met to discuss the codes allocated. Following discussion, a thematic framework was constructed, and this index was applied to the transcripts. The development and refinement of the thematic framework occurred throughout the analysis of the transcripts, with discussions of these refinements between the two members of the research team. No further themes were found towards the end of the analysis, indicating data saturation. An audit trail of the analysis process was kept through charts of initial themes and decisions regarding the refinement of the analysis process.

## 3. Results

### 3.1. Participants

Thirty people took part in the study (n = 17 PwP, n = 13 OA). Table 1 shows the demographic characteristics of the participants. The mean age of PwP was significantly lower than OA. In both groups, there were more people living with a spouse than living alone, and more people not receiving support from carers. All participants had at least one comorbidity, most commonly arthritis, joint replacements, diabetes, depression and previous cancer. These were evenly spread between the groups except for diagnosed depression, which was more prevalent in PwP. GDS scores were similar between the groups, which might suggest that depression was better diagnosed in the Parkinson’s group. The Parkinson’s specific measures are indicative of mild to moderate disease severity. Ten PwP experienced freezing of gait.

### 3.2. Gait and Balance Measures

Table 2 shows that gait speed in both groups was above the suggested indicator of frailty (0.8 m/s) [33]. Results of the Timed Up and Go (TUG) test are indicative of high falls risk in both groups, with the Parkinson’s groups being significantly higher. In contrast, while results of the Mini BESTest test agree that both groups are at high risk of falls, the OA scored significantly worse on this test. Walking aids were more commonly used by the OA.

### 3.3. Use of Activity Monitor and Body-Worn Camera

Wearing of the Axivity monitors was consistent across participants and groups, with good compliance with leaving the monitor in situ. The body-worn camera usage was more variable: one participant (OA) felt uncomfortable with the camera and, therefore, did not wear it all, and for the remaining participants, there was a mean image capture time of 10.3 (SD 3.6) hours per day. Participants were advised to remove the camera overnight, so no nighttime activity was recorded.

### 3.4. Interview Data and Camera Coding

Quotes from older adults without Parkinson’s are indicated with ‘OA’, and quotes from people with Parkinson’s are indicated with ‘PD’. Table 3 summarises the key themes and subthemes that were generated from the interview data.

Almost all participants acknowledged the importance of PA. Many felt that they were not doing sufficient levels of exercise but were unable to articulate what level they should be aiming to achieve. With the exception of one person with Parkinson’s, none of the participants were able to recall the recommended activity guidelines of 150 min of moderate PA per week [34]. When questioned, many of the respondents had no plan in place as to how they were going to increase the amount of PA they were doing, and those that did articulated extrinsic reasons for not being able to do so, for example, bad weather or ill health.

### 3.5. Priority of Physical Activities

#### 3.5.1. Past Experience of Physical Activity

For individuals who considered themselves to be active and sporty in their younger life, coming to terms with their reduced levels of PA was associated with feelings of frustration. For others, there was a sense of inevitability or resignation. Past activity was a prominent point of discussion, particularly among the PwP. Their perception of their current level of activity was strongly linked with what they had participated in in the past. There was a definite distinction between those who considered themselves to have “always been active” and those who described themselves as “not a sporty person” or “never been one for exercise”. Among those who claimed to have always been active, many attributed their longevity or good health to their level of past activity. The role of past experience of PA could serve as a facilitator or barrier to PA. Whilst those who had participated in activity in the past understood its potential benefits and aimed to adapt to their current ability, there were others for whom this was not an option. There was the opinion that if individuals could not do what they used to in relation to PA, then they did not want to do something that felt inferior or less than that.

“I can’t do what I used to in the garden, that bugs me”(OA006)

“I want to do everything…But I know inside I can’t”(PD004)

Overall, the OA expressed a desire to be doing more exercise and showed hope to engage in physical activity in the future, whereas the PwP tended to be more matter-of-fact when discussing their limitations in relation to PA. The PwP referred to the fact that they were “slowing down” as a reason for not being as active as they once were but rarely related this directly to their diagnosis of Parkinson’s.

#### 3.5.2. Activities of Daily Living and Exercise

There were no key differences between the type of daily activities that those with and without Parkinson’s were undertaking. The coded activities are shown in Table 4, with sedentary leisure activities, including watching TV, being the greatest proportion of daily activity for all participants. This was also reflected in the interviews, with illustrative codes provided. Both groups recognised that housework was a form of PA, with people striving to keep up with jobs such as cleaning and dusting, while others noted that they had always considered themselves to be houseproud but they were unable to carry out the tasks that they once did.

#### 3.5.3. Walking

Walking was an activity that was discussed by all but four participants in the study. There was a sense of regularity and routine of walking for the PwP, who mentioned the importance of getting out every day and being aware of the benefits of this on general health as well as their Parkinson’s. Very few, however, referred to the nature of their walking in relation to whether their speed, distance and length of time spent walking were sufficient to raise their resting heart rate.

Table 5 demonstrates that participants in both groups were achieving around 10,000 steps per day on average and spending around 10% of their day walking, although it should be noted that there was wide variation between individuals. The types of activities shown previously in Table 4 confirm that little of the daily walking total is accumulated through walking outdoors for the purpose of exercise but rather seems to be accumulated throughout the day, for example, during domestic tasks or moving around the inside of the house.

The subject of organised walking groups and walking with friends/spouses was touched upon by some individuals, with benefits noted around the social aspects of walking with others. However, it was the PwP who voiced concerns about not being able to keep up with other people or having to drop out of groups because they were not strong enough to keep up.

“Because I can’t do a lot of the walking like we used to, because we used to walk a hell of a lot”(PD016)

#### 3.5.4. Shopping

The routine and necessity of grocery shopping meant that for many participants, walking as part of shopping trips formed the basis of their regular exercise regimes. Within this theme, issues specific to the PD population arose in relation to anxieties around managing money and packing—these related to perceived slowness of activity and problems arising from physical symptoms such as upper limb tremor. As a result, PwP often mentioned the need for friend, family or staff support at the supermarket, specifically when reaching the tills. The need for physical support to negotiate the supermarket isles was more of an issue for the dependent OA.

“I mean I’m responsible for the shopping, right. I do it every Thursday”(PD007)

“I enjoyed the bit shopping…Oh I’m…that’s the trouble. I’m not steady. But yes, if I went with [companion] well I could hang on to him…So I bought myself some bits”(OA010)

#### 3.5.5. Differing Purposes and Values Related to Physical Activity

The value that participants placed on exercise was dependent on their expectation of what the purpose of this exercise was. For those who just wanted to leave the house, their expectation was that this would allow them to keep going and maintain current activity levels. However, for others, there was a strong sense that exercise would make them better.

“If you don’t move you’ll lose it, you know”(OA006)

“One of the things I was hoping, that I could train my way out of it. I was hoping that by doing this thing at the pool I’d be able to get back to where I was, and I can’t”(PD017)

For those OAs that were undertaking regular PA, a number talked about the need to push (or ‘force’) themselves to be more active. They mentioned setting themselves personal challenges or projects to make them work harder or walk faster. This was less apparent amongst the PwP

“…it’s very easy to not bother to go…out in the evening for example”(PD003)

The concept of what constituted exercise differed amongst the participants, largely dependent on their level and involvement in activity in the past as well as their current state of health. For some, ‘anything and everything’ was considered PA: getting up and down in the house, getting in and out of bed, vacuuming or simply walking around. For others, the concept of PA and exercise was based on attendance at a gym or participation in sports.

### 3.6. Influence of the Environment

#### 3.6.1. Safe at Home

For some respondents, there was a definite sense of their house being a safety zone. Table 6 shows that for both groups, a significant proportion of the week was spent at home, which aligns with the common activity types seen in Table 4. Both groups talked about feeling more secure and more confident in their own homes. Part of this was related to the environment, with priorities placed on flat walking surfaces and things to hold on to; many reported not needing their walking aids indoors due to the use of furniture or walls for support.

“I’m more safe in there than I am anywhere else in the house… it’s small and most confined area of the house”(PD003)

“But not in and around the house because I’m in my safety zone here”(OA005)

#### 3.6.2. Weather

Given that the majority of participants regarded walking to be their main source of PA, the effects of seasonal weather changes are an important area for consideration when discussing barriers to participation, and they were frequently discussed. For those who were interviewed during the winter months, snow and ice causing slippery conditions caused significant anxiety.

“But I was a bit nervous about going to the class during the winter because it gets very icy out here…And I get nervous about walking when it’s icy…And just the pavements get so slippery”(PD001)

Poor weather conditions were stated as reasons for not venturing out of the house or for activity avoidance. Both groups expressed concerns about reduced balance and gait abnormalities. They made reference to insufficient step height and/or length as weaknesses in their walking ability, and this was closely linked to the surface on which they were moving. A large number of respondents cited the state of the pavements as a limiting factor on the quality of their walking outside, with concerns about uneven paving resulting in shuffling gait and walking with their heads down, which was not specific to those with Parkinson’s.

“…all the flagstones going up…are badly placed…And I tend to walk with my head down watching where I’m putting my feet”(PD009)

“…as soon as I hit the pavement idea the shuffle becomes more distinct”(OA006)

#### 3.6.3. ‘Getting out of the House’ and ‘Keeping Going’

In contrast to those individuals discussing safety at home, particularly for the PwP, the desire to “get out the house” was as strong a motivation to remain active because of the physical benefits of PA, although participants did refer to factors such as not losing power in their legs and becoming steadier on their feet. There was a feeling that participants felt they had to or needed to be doing something but were often unable to articulate what that was specifically. However, being out of the house was deemed important.

“…the gym or something like that, that keeps you going as well, I think”(PD005)

“I just feel as though I’ve got to be out doing something”(PD016)

#### 3.6.4. Falls

At the forefront of a number of individuals’ minds is the concern of falling again, and the individuals are worried about this. The participants often struggled to describe a clear cause of their fall, but they did allude to various activities that made them feel vulnerable since falling. Only one participant (PwP) experienced a fall during the recording period. The camera captured images of the fall occurring while preparing a drink in the kitchen, which involved turning in a small space while dual-tasking. The participant fell forwards to the floor, resulting in a facial injury. Axivity data showed a short period of sitting following the fall and then a return to the usual pattern for that person. When discussing these images in the interview, the person was not able to describe what had caused the fall, just describing themselves as being ‘on my feet one minute and on the floor the next’; they expressed this being quite normal for them and had not felt they had made any changes to their activity for the rest of the week, but did report already having modified some activities, such as walking outdoors, due to the risk of falls.

Going outdoors was often perceived as risky, with a fear of being outdoors, especially when alone, as reflected in Table 6 by the small proportion of the week spent outdoors.

The issue of using stairs was particularly prevalent amongst the OAs. Participants were very wary of the stairs, whether they had had a trip or fall on them or not. Some opted to only use the stairs once a day for bed, whilst others talked about having to take extra care and not taking any chances when negotiating them. The OAs also referred to their use of bannisters and rails to assist them on the stairs as well as techniques such as non-reciprocal stepping and backwards descent. The issue of knees/legs giving way was mentioned by several OAs as a factor in their fear of the stairs.

Several of the PwP described trips or falls they had had on the stairs. Unlike with the OA, none of these falls resulted in those affected showing any fear avoidance behaviours around the use of the stairs.

“And I missed the bottom step and fell into this bannister here and the wall here and I hit my head”(OA005)

“I remember. I did. I came down the stairs one morning… [fell] By the radiator…Yeah. I think one of my knees gave away”(PD016)

The physical impact of falls, including broken ribs and stitches in the head, was acknowledged but not a large focus in the interviews. Whilst freezing is mentioned by some participants in the PD group, the participants tended not to focus very much of their discussion on Parkinson’s symptoms.

“Aye… I’ve fallen a lot, but I don’t tell anyone [laughs]… I mean, he often hears us thud, you know—I’ve had black eyes, and all sorts, you know, bruises up me arms… It’s terrible…”(PD001)

### 3.7. Physical Limitations and Their Management

#### 3.7.1. Physical Limitations and Physical Deterioration

Participants from both groups related broader health issues to PA, with lower limb musculoskeletal complaints frequently cited as limiting participation. Other illnesses such as previous cancer diagnosis and treatment and cardiovascular disease also featured as barriers, with reference to prolonged hospital stays and post-operative recovery times contributing to reduced levels of fitness and participation.

“Just my knees. I wish I could be more active…You know, I wish…I wish it didn’t hold us back”(OA005)

“I think it was the breast cancer slows you down. Well an operation takes some of your strength away doesn’t it?”(PD004)

#### 3.7.2. Perceptions of Aging

The OA group was significantly older than the PwP (Table 1), but both groups referred to aging as an influence on PA. For some, there was a strong sense of negativity around the ageing process, whilst others were more accepting of what they perceived to perhaps be an inevitable decline in function with age. Reduced motivation to exercise was a theme that emerged in relation to ageing for both groups, whilst it was mainly the OA who made reference to the effect of an ageing social network on their PA levels (although this may be attributable to the older demographic of this group, see Table 1).

There was a definite sense that some participants dissociated themselves from certain activities that they deemed to be for “younger people”. This may be linked to self-efficacy, as well as social norms and expectations:

“these er gyms that you talk of er…they’re for younger people that are trying to, you know, impress some…with their muscle at the ready. I’m not too bothered”(OA007)

“So I had this thing, I’ve never said this to [wife]…that I might sort of keep going till I’m 80 and then call it a day”(PD007)

#### 3.7.3. Management of Physical Problems

Employing strategies such as taking care and walking slowly to negate the risk of falling was a common finding. Other strategies to reduce the likelihood of future falls included home modifications, standing up slowly, thinking about walking, walking with head down and slowing down.

A number of participants reported the use of a walking aid such as a stick or walker (Table 2). In contrast, some individuals discussed feeling too self-conscious to use a walking aid. Holding on to furniture/surroundings was reported by some individuals. Five OA used a walking stick, and four used a wheeled walking frame, while the remaining four used no walking aid (Table 2) but did report requiring assistance from another person at times. Fewer PwP used a walking aid (four used a walking stick, two used a wheeled walking frame and eleven used no aid). Wheelchairs were mentioned, but one individual did not want to use this yet, and another acknowledged that, whilst it was easier, it restricted the amount of exercise they would get:

“I walk with my stick…If I was to walk into the middle I would be all over the shop…So I take my stick with me wherever I go”(OA006)

### 3.8. Psychological Factors

#### 3.8.1. Anxiety Associated with Mobility and Falls

There was no difference between groups in the falls efficacy scale, with both groups scoring in the ‘high concern’ bracket (Table 2). People with PD noted the impact of falls on their confidence, stating that concern over falling again limited their daily PA. Uncertainty regarding the cause of falls contributed to fear, with participants commenting on the uncertainty surrounding why they fell. There were similar findings within the OA group, with some individuals reporting being concerned and fearful due to their previous fall and feeling unable to go out due to the risk. Similarly to the PD group, the reported taking more time foe these tasks. 

“It has [affected my activity levels] because I very rarely go out on my own now… or walk anywhere on my own. It has left me feeling [pause] rather nervous”(PD005)

#### 3.8.2. Confidence and Embarrassment

Feeling self-conscious was discussed among some participants in both groups. Some PwP described increased levels of anxiety as a trigger for their Parkinsonian symptoms. For example, trips to the supermarket were particularly challenging because they felt under pressure to move through the checkout quickly. Knowing they could not do this, the anxiety associated with the fear of holding people up subsequently worsened their PD symptoms.

“I used to be in a bit of a knot because I’d try to get the money out and I was conscious of the tremor, and trying to get the shopping and the change and the- and I always used to get in a panic then”(PD005)

In contrast, there is a strong sense in several PwP of not letting this restrict activities they wish to undertake, from household tasks to skydives. Individuals highlighted they were conducting activities despite Parkinson’s disease and putting this to the back of their minds. “I’ve done things that I wouldn’t have done possibly, if I hadn’t had Parkinson’s” (PD005).

The psychological impact of not being able to keep up or requiring extra help was mentioned in relation to gym-based exercise. There were concerns regarding gym equipment use, feeling self-conscious and being embarrassed. Whilst exploring the concept of stigma, as it manifested within the interviews, there was a strong sense of some PwP that their negative self-perception in relation to their physical symptoms impacted their confidence to go out in public.

“but when you get older you think everybody’s looking at you…oh have you seen that old wife on that bike”(OA010)

”But like last night I was twitching like mad and I said…I’m not going anywhere. I’m not having people looking at us”(PD014)

## 4. Discussion

The aim of this study was to explore the types of activities and the time spent in different environments of people with and without Parkinson’s who have fallen. The results showed similar levels of activity between the groups. While the number of steps per day was relatively promising, time spent doing structured exercise and time spent out of home was low, with both groups citing numerous barriers to being active, some of which are directly related to falls.

Both groups showed high concern around falling, as measured with the FES-I and through qualitative comments describing an awareness of falls risk, avoidance of certain activities and preference for the home environment for most participants, in agreement with previous qualitative and quantitative research [11,35]. While freezing of gait was discussed by some PwP, many of the reasons for adapting activity or not being as active as they would like to be or feel the need to be were shared between the groups.

Participants acknowledged the impact of various health conditions, including Parkinson’s, on their PA levels, particularly in relation to the resulting effect on social disengagement and in agreement with previous studies [17,36,37,38]. Participants described physical limitations, difficulty with accessing different environments, lack of confidence and motivation, and also the effects of stigma around ageing and Parkinson’s as significant influences on their engagement in PA and exercise, particularly within a group or public settings. The issue of ‘slowing down’ was something that influenced both OA and PwP in their decision-making when going out in public, as well as their participation within group settings.

The average step count per day was around 10,000 for both groups, which is high compared to the average step count measured in previous studies for older adults [39] and PwP [40,41], and the average bout length was around 14 s for both groups. This indicates that the stepping activity was accumulated in very regular but very short bursts of activity, with the location codes demonstrating that the vast majority of this activity took place at home. Through the interviews, the participants in both groups described the importance of keeping active through daily tasks, such as housework or simply avoiding sitting too long. While it is promising that the participants appeared to be avoiding sedentary behaviour with regular short bouts or moving, there is a clear opportunity for building from this to encourage more structured exercise [15,16,42]. While the use of the Axivity monitor is well validated for measuring free-living activity, including step count in this population [13], we cannot discount the possibility that wearing the camera artificially increased the step count in this study when considering the relatively high daily step count in both groups. While this was not supported in the interviews, where participants reported feeling their activity was reflective of their normal, we did not measure any step counts without the camera being worn to provide an objective comparison.

While both group’s performance on the balance tests indicated a risk of falls, there were differences between the groups. PwP performed worse on the TUG test, while OA performed worse on the Mini BESTest test; this is perhaps explained by the TUG being more impacted by the slow movement initiation and turning for the PwP [43], and was perhaps influenced by the test being conducted in the home environment [44], although we do not have clinic/lab-based measures to confirm. It should also be noted that in our study, the OA group was significantly older (mean age of 81.8 years, compared to 72.5 years for PwP), and age is associated with poorer performance on the Mini BESTest test [45].

It was clear within the interviews that most participants were unclear about what constitutes physical activity and, therefore, what they should be aiming for; this is in agreement with previous studies regarding awareness of physical activity guidelines in older adults [46]. Much of the guidance for professionals working with older adults and those with Parkinson’s emphasizes the importance of educating people about the benefits of physical activity and exercise [13,47], but further work is needed to develop effective methods of sharing this message.

While most participants reported being aware of the importance of being active, there was a dichotomy in some of the interview responses relating to future plans, with some expressing a desire to increase their level of activity while others described feeling this was not a priority at their age. None of the participants indicated aiming for any particular level of intensity, frequency or duration of activity, which may make objectively setting targets more difficult. This is something that professionals providing advice should consider, providing tools for monitoring and, where appropriate, progressing activity [47,48]. Shared decision-making and collaborative goal-setting should be an integral part of supporting older adults living with long-term conditions [49], but this requires a clear understanding of the reasons for and ways of achieving any intervention.

While guidance for both reducing the falls risk and managing Parkinson’s emphasises the importance of exercise [50], it appears that most of the participants found it challenging to engage with any kind of structured exercise programme. While some people described walking regularly and others did have an exercise programme, most were aware of the benefits of exercise but were unclear on how to navigate various barriers, including physical limitations associated with multimorbidities, concern about being outside or attending exercise venues and fear of stigma.

The participants in our study referred to several strategies used to remain active while minimizing the falls risk, such as using walking aids, being vigilant when walking and engaging in what they perceived as safe forms of exercise, such as the gym, rather than outdoor activities. It is important that professionals providing exercise advice take these preferences into account; utilizing strategies that the person already feels are realistic and helpful is an important aspect of shared decision-making [14,15,42,48]. Understanding the rationale behind the chosen activity or avoidance strategy will allow the professional to educate on the risk of falls as well as the risks associated with reduced activity in a personalized and tailored way [14,17]. Some very pragmatic issues, such as seasonal weather conditions, which were highlighted by several of the participants, should be taken into account with help to find alternatives when needed.

As we did not include a control group of people who had not fallen, it is not possible to discuss in any detail whether the behaviour of our participants differed from non-fallers. However, there are suggestions from our participants that their activities have changed since their falls, for example, using mobility aids or avoiding being outdoors when alone. This shows that falls do impact daily routines and activity levels, which is important to acknowledge. It may be that some adaptations following a fall are potentially harmful, such as reducing the amount of time spent standing or moving or avoiding leaving the home, which could lead to isolation. It is, therefore, important for professionals to explore these behaviors in order to provide education and support and to find solutions that allow the person to maintain their activities while feeling safe.

This emphasizes the importance of a personalised approach when encouraging physical activity and exercise [15]. Health professionals supporting people with Parkinson’s or older adults who have fallen need to acknowledge that knowledge of the benefits of physical activity and exercise is not enough to fully address sedentary behaviour. Influences on activity levels are complex, and these must be taken into account in order to provide options that are achievable, accessible and desirable to the person.

### Strengths and Limitations

Some important limitations need to be acknowledged in relation to this study. While body-worn cameras provide a different perspective and important contextual information not possible to capture using other methods of wearable technology, they create some not-insignificant challenges for both the researcher and the participant. While compliance and wear time were generally good, for all participants, the camera was deliberately or accidentally obscured for significant portions of the day, meaning a full picture of habitual activity was not collected. Cameras are also more obtrusive than other forms of wearable monitors, meaning we cannot discount the possibility that wearing the camera influenced the participants’ behaviour. It is also possible that the use of the camera and Axivity monitor may have positively influenced participants’ levels of physical activity and exercise, which could go some way towards explaining the unexpectedly high step counts that were recorded by individuals during the study period.

In addition, the groups with and without Parkinson’s were not matched for age, with the Parkinson’s group being significantly younger. Experiences may change for people living with Parkinson’s as they age, and we may have found different qualitative and quantitative findings with an older sample. The sample size of the current study did not allow for sub-group analysis, which may be useful in future mixed methods studies to explore differences in experiences and perceptions, for example, by disease severity or age.

There was no specific measure of anxiety used in this study, which may be an important omission given the link to both falls and physical activity. While the FES scores indicate a high level of concern regarding falling, it may also be helpful to understand the influence of anxiety more generally in future research. As participants highlighted increased concern around mobility and falls during the winter months, it would be appropriate in future and larger studies to explore any impact of seasons on the quantitative measures, and longitudinal interviews would be interesting to explore this further.

The interviewer (J.D.) is a registered physiotherapist with clinical expertise in Parkinson’s, which may have led to better iterative questioning and credibility. However, while all participants were encouraged to be open and honest during the interviews, an existing researcher–participant relationship between the interviewer and participants may have affected open dialogue and participant responses. Every attempt was made to minimise this, and the use of the camera images aided focused discussions. The interviewer also completed a reflective diary between interviews.

This paper has not addressed the social aspects of physical activity and falls in any detail. Body-worn camera coding was used to indicate the proportion of the day spent in the company of others, and this was a theme explored in the interviews. Due to the complexity of the issues raised, this will be reported separately.

## Figures and Tables

**Figure 1 geriatrics-10-00008-f001:**
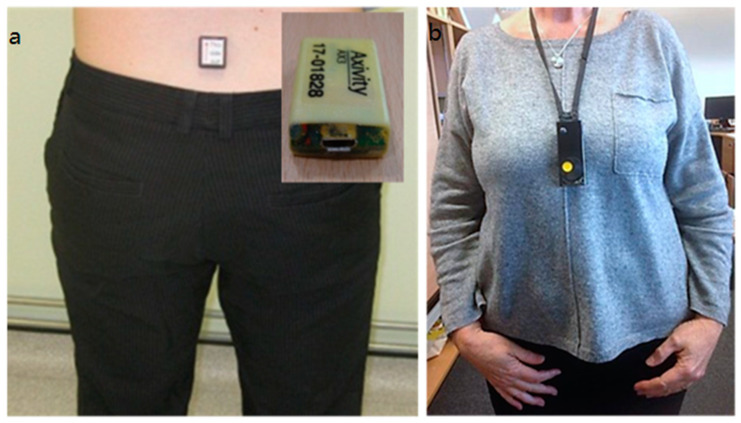
(**a**) Axivity monitor situated on the lower back and (**b**) body-worn camera.

**Table 1 geriatrics-10-00008-t001:** Demographic characteristics of participants with and without Parkinson’s.

	Older Adults Without Parkinson’s (OA)	People with Parkinson’s (PwP)
n	13	17
Gender F/M	6/7	7/10
Age, years Mean (SD)	81.8 (8.1) *	72.5 (5.1) *
Living situation n (% of group)	6 (46%) living alone, 7 (54%) living with spouse	5 living alone (29%), 12 (71%) living with spouse
Support from carers n (% of group)	None: 8 (62%) Informal (family/friends): 3 (23%) Formal (paid) domestic: 1 (7.5%) Formal (paid) personal care: (7.5%)	None: 12 (70%) Informal (family/friends): 2 (12%) Formal (paid) domestic: 2 (12%) Formal (paid) personal care: 1 (6%)
Geriatric depression scale † n (% of group)	No depression: 7 (54%) Mild: 3 (23%) Moderate: 3 (23%) Severe: 0 Median = 3	No depression: 9 (53%) Mild: 5 (29%) Moderate: 2 (12%) Severe: 1 (6%) Median = 4
Hoehn and Yahr Median (IQR)	N/A	3 (0.5)
Time since diagnosis, years Mean (SD)	N/A	15.78 (10.12)
Unified Parkinson’s disease rating scale, UPDRS, motor part 3 Median (IQR)	N/A	24.01 (21.5)
Freezing of gait questionnaire Median (IQR)	N/A	10 freezers 21.0 (11.5)

* Indicates a significant difference between groups; N/A—not applicable, Parkinson’s specific outcome measure. † Score of 0–4 means no depression, 5–8 mild depression, 9–11 moderate depression and 12–15 severe depression.

**Table 2 geriatrics-10-00008-t002:** Gait and balance measures of participants with and Without Parkinson’s.

	Older Adults Without Parkinson’s (OA)	People with Parkinson’s (PwP)
Gait speed (m/s) Mean (SD)	1.9 (1.1)	2.0 (2.8)
Timed Up and Go (s) (low score suggests better performance) Mean (SD)	19.3 (10.3) *	28.6 (67.4) *
Mini BESTest (high score suggest better performance) Median (IQR)	10 (7.5) *	17.0 (10.5) *
Falls efficacy scale Median (IQR)	37 (21.5)	39 (24.5)
Use of walking aids n (% of group)	None: 4 (31%) Stick: 5 (38%) Wheeled walking frame: 4 (31%)	None: 11 (65%) Stick: 4 (23%) Wheeled walking frame: 2 (12%)

* Indicates a significant difference between groups.

**Table 3 geriatrics-10-00008-t003:** Key themes and subthemes.

Key Themes	Subthemes
Priority of physical activities	Past experience of physical activity Activities of daily living and exercise Walking Shopping Differing purposes and values relating to physical activity
Influence of the environment	Safe at home Weather Getting out of the house and keeping going Falls
Physical limitations and management	Physical limitation and physical deterioration Perceptions of aging Management of physical problems
Psychological factors	Anxiety associated with mobility and falls Confidence/embarrassment

**Table 4 geriatrics-10-00008-t004:** Types of activities captured on body-worn cameras. No significant differences between groups.

	Older Adults Without Parkinson’s (OA)		People with Parkinson’s (PwP)	
	Mean % of Captured Time	SD	Mean % of Captured Time	SD
Sedentary leisure (computer/tablet/phone, reading, craft and games)	21.36	8.32	34.08	7.88
Watching TV	18.42	13.23	18.99	17.81
Mobility inside house	11.21	1.45	19.16	2.40
Domestic tasks	9.32	1.07	11.49	1.25
Eating and drinking		1.89	5.79	2.93
Talking	3.63	4.56	7.01	7.15
Self-care (washing, dressing, taking medication)	1.77	0.54	2.62	1.06
Admin and paperwork (mail etc.)	1.01	0.84	2.87	1.97
With children	0.05	0.12	1.32	1.29
With pets	0.00	0.00	0.94	1.48
Resting	3.47	9.61	3.89	5.84
With carer	0.08		0.00	
Transport (driving, passenger, public transport)	14.43	2.60	12.04	2.09
Walking outside	2.58	2.23	3.94	3.74
Activities outside of house (hairdresser, cinema, restaurant)	2.47	0.34	2.33	0.13
Gardening/DIY	3.12	1.79	3.60	1.47
Dancing	0.53		1.48	1.34
Shopping	0	0	3.35	1.69
Illustrative quotes	“I haven’t got much else to do—bet a couple of horses, put the television on—I’m a horsey fan you know” (OA001) “And the afternoon is watching the telly again and it’s sport that I watch more than anything else…Passes the day” (OA007) “I like reading” (OA006)		“I just get my nose in a book….Or a puzzle or I started doing some knitting” (PD002) “I tend to have my meals…And then erm I watch a bit television…Then that’s about it really” (PD009) “I’m always on the computer” (PD013)	

**Table 5 geriatrics-10-00008-t005:** Comparison of Axivity data between groups.

	Older Adults Without Parkinson’s (OA) Mean (SD)	People with Parkinson’s (PwP) Mean (SD)
Total walk time (h)	16.84 (6.72)	16.74 (7.21)
Total walk time per day (min)	144.37 (57.62)	151.75 (56.48)
Steps per day	9825.04 (4033.78)	10,374.73 (4051.69)
Mean bout length (s)	14.71 (3.52)	14.73 (3.00)
Percentage of walking time per day	10.03 (4.00)	10.54 (3.92)
Illustrative quotes	“I’ll walk around the shops and I’ll walk around the corner, walk to the bus” (OA013) “We’ll walk through the shopping centre, right the way back round” (OA005)	“Well, of course, if you don’t go walking you’d take no exercise at all” (PD007) “Well, I think its active just cleaning the house…and changing the bed upstairs” (PD004)

**Table 6 geriatrics-10-00008-t006:** Percentage time indoors and outdoors and qualitative quotes relating to weather and safety at home.

	Older Adults (OA) Mean (SD)	People with Parkinsons’s (PwP) Mean (SD)
Indoors—home (% of captured time)	78.6 (14.1)	79.6 (13.7)
Indoors—not at home (% of captured time)	9.8 (9.1)	10.5 (6.9)
Outdoors (% of captured time)	8.6 (8.2)	10.1 (7.6)
In vehicle (% of captured time)	5.0 (3.1)	6.3 (7.6)
Illustrative quotes	“So I keep saying when the weather gets a bit warmer [laughs] I’ll do it and I think that’s silly because I am putting it off-” (OA006) “I would like to go to the baths, but it’s still too cold for me to go” (OA010)	“The pavements are bad” (PD011) “We walk when we can don’t we. If we get decent weather we’ll go out” (PD016)

## Data Availability

Data are unavailable due to privacy or ethical restrictions; this was not granted in the ethical approval for this study.

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
