# Peer review of "Exploring Context Allows Us to Better Understand Physical Activity in People with and Without Parkinson’s Who Have Fallen: A Mixed Methods Study"

_geriatrics, 2025, doi:10.3390/geriatrics10010008_

Round 1

Reviewer 1 Report

Comments and Suggestions for Authors

This manuscript addresses the timely research question of the types of activity people with Parkinson’s disease (PwP) and older adults without Parkinson’s disease (OA) who had fallen in the past year, engaged in. The combination of a qualitative and quantitative analysis is informative in addressing this question and gives valuable in the activities of the study population.

I have outlined my comments and suggestions below. 

The aim and the implications of this study are mentioned in the introduction: 

“Having more insight into the types of activities PwP engage with and the environments in which they spend time, will allow us to better tailor our advice around maintaining PA while reducing falls risk. The aim of this study, therefore, was to explore the types of activities and environments of PwP who have fallen to identify barriers and facilitators to PA.” and“…. This creates a tension for professionals who promote exercise and remaining physically active as an important factor in managing motor and non-motor complications of Parkinson’s disease.”

·   This study mentions several strategies participants use to remain active while minimizing fall risks, such as careful walking, using walking aids, and engaging in safe forms of exercise such as gym instead of outdoor activities​. Nevertheless, the connection between these strategies and specific advice for maintaining PA while reducing fall risks could be more explicit. The practical application of study findings in providing tailored advice is not sufficiently detailed. Can the authors draw any conclusions based on the results? If not, what are the scientific future recommendations to address this issue? 

·   The focus of the interview does not give us any insight into the causes of falling. The actual situations in which participants have fallen (apart from on the stairs) is not discussed, which makes the interpretation of the barriers and facilitators harder to interpret. Whether or not the current behaviours of those who have had a fall in the past year contribute to fall risk, remains elusive. 

This is an example of avoidance behaviour related to falling: “At the forefront of a number of individuals minds is the concern of falling again and individuals being worried about this. Going outdoors was often perceived riskier than indoors and a fear of outdoors and feeling nervous of going out alone reflected in table 6 by the small proportion of the week spent outdoors.” But this is not echoed in regard to taking the stairs in PwP: “none of these falls resulted in those affected showing any fear avoidance behaviours around use of the stairs.” 

·   Did any of the participants fall during the study? Did their behaviour (on wearables) differ before and after? 

·   This study does not include any control group of people who have not fallen, which would be helpful to compare if fallers exhibit different kinds of behaviour. The paper does include some discussion on how participants manage their activities post-fall, with strategies such as modifying home environments and using mobility aids​. However, a discussion around behaviours before and after experiencing falls is not thoroughly explored either. This comparison is relevant because it can provide insights into how falls impact daily routines and activity levels, and what specific changes or adaptations are made to mitigate future fall risks.

·   The combination of qualitative and quantatiative methods is very informativie. This study could enhance its findings by comparing these outcomes. For example, taking into account the severity of Parkinson's disease, or their activity levels measured with the wearables when participants describe their behaviour can provide a more nuanced understanding of their experiences. This mixed methods approach can help identify specific needs and tailor interventions more effectively.

Some minor points that can use some clarification: 

·       For better context, could you provide the cut-off scores for the GDS and the mean (or median) and range? 

·       It would be helpful to know if there was any specific measure of anxiety, given its relevance to fall risk and activity levels."

·       Since some participants mentioned increased anxiety during winter months, it would be useful to understand if and how the season was accounted for in the analysis.

·   “All of the OA used a form of mobility aid or assistance at times.” This seems to be in contrasts with the results in Table 2, where 31% of older adults reported not using any walking aids​. 

·   “In contrast while results of the Mini BESTest agree that both groups are at high risk of falls, the OA scored significantly worse on this test, this could be indicative of the slow movement initiation for the PwP impacting on the time taken to complete the Timed Up and Go test.”  Could you clarify the interpretation of the Mini BESTest results in relation to the Timed Up and Go test? This would help in understanding the link between balance and movement initiation in PwP.

Author Response

Many thanks for your helpful review of the manuscript. 

Please see attached document which summarises our responses to reviewers comments.

Reviewer 2 Report

Comments and Suggestions for Authors

This study lifts and important topic and in theory the application of mixed methods to investigate physical activity patterns among fallers is relevant. The use of the body-worn cameras is novel and a strong merit for the paper in terms of originality. 

Executing a well-designed mixed methods paper is however a challenge, often harder than sticking to one of the chosen methods. This paper, in it's present form, doesn't reflect enough scientific rigor in terms of methodology/ design. For example I am left with the following questions:

-What was the nature of mixed methods design? (See Creswell for guidance).

- When/ and how were the different data forms merged, for example at what stage, and why so?

- The Qual and Quant data are presented alongside each other, but I don't see any real merging regarding interpretation. 

- What was the aim of the qualitative interviews? I lack the interview guide.

- Was the TUG test a 3 meter walk? If so I find the completion times very long for the PD group who were H & Y 3. Also 10 000 steps/ day is significantly over the average for this group of moderatley progressed. It makes me questions the methods. Maybe unecccessarily so (I don't see an adequate discussion of this either)

- The Qual findings incorporate some interesting insights. But I see them as more of a first draft of qualitative analysis. They are presented over 6 pages, which is excessive, even if this was soley a Qual paper. The Qual results prob reflects that the analysis need more interpretation and condensation and merging with the quant data. 

- This discussion is not either developed, far to short, lacks a grounding in the literature and after 6 pages of Qual, results is also not worked through.

In summary- these results have merit and are not done justice in the phase. 

I would go back to the drawing boards and consider a quant or Qual study, of a radical increase in methodological stringency and re-structuring of the results.  

Author Response

(The authors gave the same response as above.)

Round 2

Reviewer 1 Report

Comments and Suggestions for Authors

Due to the nature of the study, not all concerns could be addressed but they are adequately discussed in the limitation sections. I would recommend this paper for publication.

Author Response

Thank you for your comments. 

Reviewer 2 Report

Comments and Suggestions for Authors

There are some improvements in terms of merging the data. One of my main concerns remains however.

I lack description (in the Background) or any comparison of current findings (in the discussion) with previous qualitative studies which have investigate Falls or physical activity perceptions in PD (or among older adults in general). This would be standard in my view for a study which has chosen to investigate perceptions of falls and physical activity.

The newly added text to the results (line 249-252) is strangely placed it feels like a discussion, and speculation. Perhaps the tug was influenced by it being performed at home, perhaps not, maybe the test wasn’t applied correctly, all number of reasons? This text is more of a discussion, regardless.

In contrast 247 while results of the Mini BESTest agree that both groups are at high risk of falls, the OA 248 scored significantly worse on this test. The reason for this difference (7 of 25 worse on TUG, OA performing worse on Mini BESTest) is perhaps explained by the TUG 250 being more impacted by the slow movement initiation and turning for the PwP, and was 251 perhaps influenced by the test being conducted in the home environment

Here are some:

The qualitative result section is still far too long. No further condensation of the qualitative findings have occurred, from what I can see, and that was my major concern (still 6 pages long). This reflects that they are premature in my analysis and need further work. Interesting findings get lost, too many citations. Requires further qualitative interpretation and abstraction which always results in a condensation of the text.

Discussion

The high step count observed needs also to compared to the bulk of the Pd literature reporting step counts that are lower, in a lot of cases an average of 5000 in cohorts with faster walking speeds.

Ref 34 (barriers and facilitators) is one study, needs addition of more, both  qual and perhaps quant.

Comments on the Quality of English Language

The formulation need to be made more concise throughout. More time needs to be spend on formulation and condensation of the 23 page text. It requires too much of the reader. 

Author Response

Thank you for comments, please refer to the attached to see our response to each point. 
